# The Double Dyson Index *β* Effect in Non-Hermitian Tridiagonal Matrices

**DOI:** 10.3390/e25060868

**Published:** 2023-05-29

**Authors:** Cleverson A. Goulart, Mauricio P. Pato

**Affiliations:** Instituto de Física, Universidade de São Paulo, Caixa Postal 66318, São Paulo 05314-970, SP, Brazil; cleversonagoulart@gmail.com

**Keywords:** random matrix theory, *β*-ensembles, pseudo-Hermitian, PT-symmetry

## Abstract

The Dyson index, β, plays an essential role in random matrix theory, as it labels the so-called “three-fold way” that refers to the symmetries satisfied by ensembles under unitary transformations. As is known, its 1, 2, and 4 values denote the orthogonal, unitary, and symplectic classes, whose matrix elements are real, complex, and quaternion numbers, respectively. It functions, therefore, as a measure of the number of independent non-diagonal variables. On the other hand, in the case of β ensembles, which represent the tridiagonal form of the theory, it can assume any real positive value, thus losing that function. Our purpose, however, is to show that, when the Hermitian condition of the real matrices generated with a given value of β is removed, and, as a consequence, the number of non-diagonal independent variables doubles, non-Hermitian matrices exist that asymptotically behave as if they had been generated with a value 2β. Therefore, it is as if the β index were, in this way, again operative. It is shown that this effect happens for the three tridiagonal ensembles, namely, the β–Hermite, the β–Laguerre, and the β–Jacobi ensembles.

## 1. Introduction

The possibility of reducing a full matrix into a tridiagonal matrix by employing unitary transformations inspired some researchers [1] to search for ensembles of tridiagonal matrices that would have the same properties of the ensembles of random matrix theory (RMT) [2]. This objective has been achieved in a seminal paper [3] in which ensembles of tridiagonal Hermitian matrices are constructed for the classes associated with the Hermite and the Laguerre polynomials. The tridiagonal matrices associated with the Jacobi polynomials, on the other hand, were obtained in [4,5,6]. In these three tridiagonal ensembles, the Dyson index, β, which in RMT has the integer values 1,2, and 4, can assume any positive real value. Therefore, the β ensemble generalized Dyson’s “three-fold way” [7] associated with invariance under the orthogonal (GOE), unitary (GUE), and symplectic (GSE) transformations of the three classes of the Gaussian ensemble.

Once the β ensembles were established, it was natural to investigate what happens if its Hermitian condition is removed. Of course, this interest parallels the introduction of the non-Hermitian random matrices by Ginibre [8] a few years after Wigner’s proposal of the Hermitian random matrices by the end of the 1950s [9]. In the tridiagonal Hermite case, this investigation was undertaken in ref. [10], in which, besides the question of introducing non-Hermiticity in β ensembles, there was also another motivation.

In fact, the main interest of this study was in the special class of non-Hermitian operators in which the operator is connected to its adjoint by a similarity transformation; namely, it satisfies the condition
(1)$$A^{†}=\eta A\eta^{-1}$$
where *A* is non-Hermitian and η is Hermitian. This condition defines a pseudo-Hermitian operator [11], and it implies that its spectrum is made of real or complex conjugate eigenvalues. This kind of non-Hermitian operator attracted a lot of interest with the discovery that Hamiltonian invariants under the combined parity and time-reversal transformation, called *PT* symmetry, also have real or complex conjugate spectra [12,13,14]. This led to an extension of quantum mechanics to include this special class of non-Hermitian Hamiltonians with *PT* symmetry [14] (see [15] for a review).

Concomitantly, there have been attempts to find, in the context of random matrix theory (RMT), ensembles of matrices satisfying the above relationship [16,17]. In [10], it was shown that in its tridiagonal form, the removal of the Hermitian condition produces matrices that satisfy the above pseudo-Hermitian constraint. We are here revisiting that work to report an important result that has since been overlooked. In fact, there, it was shown that the eigenvalues of the non-Hermitian ensemble are real and distributed with the same semi-circle law as those of the Hermitian ensemble. However, not much could be said about the fluctuations. In this respect, what we have now found is that asymptotically, that is, for arbitrary large matrices, the passage to non-Hermiticity is accompanied by the doubling of Dyson’s β index. It is important to emphasize that, in order for this doubling index effect to occur, it is necessary to take into account the presence of traces in the structure of the joint distributions of matrix elements in the case of the Hermite and Laguerre ensembles. In the Jacobi ensemble, it will be shown that the important quantity is the variance of the diagonal and the off-diagonal elements. Therefore, the effect occurs in the three β ensembles. Moreover, it affects the short-range statistics, measured by the nearest-neighbor spacing distribution (NNSD), and the long-range statistics, measured by the number variance. This is illustrated with plots of NNSD for the three ensembles and number variance for the Jacobi ensemble. The figures with similar results for the two other ensembles are omitted.

## 2. Preliminares

We start by recalling some results from [10]. Considering the general case of a non-Hermitian tridiagonal matrix, *A*, with a diagonal $a=(a_{n},\ldots ,a_{1}),$ an upper sub-diagonal $b=(b_{n-1},\ldots ,b_{1})$, and a lower sub-diagonal $c=(c_{n-1},\ldots ,c_{1}),$ in which elements can assume any real value, it is easily proved that the matrix *A* is pseudo-Hermitian. In fact, by defining the diagonal matrix η, whose elements are given by
(2)$$\mathrm{diag}\left(\eta \right)=\left(1,\frac{b_{n-1}}{c_{n-1}},\frac{b_{n-1}b_{n-2}}{c_{n-1}c_{n-2}},\ldots ,\frac{b_{n-1}b_{n-2}\ldots b_{1}}{c_{n-1}c_{n-2}\ldots c_{1}}\right),$$
it is immediately verified that *A* and its adjoint $A^{†}$ satisfy Equation (Equation 1), belonging, therefore, to the class of pseudo-Hermitian matrices. Moreover, by defining the diagonal matrix $\sqrt{\eta }$ whose elements are obtained by taking the square roots of the elements of η, that is,
(3)$$\mathrm{diag}\left(\sqrt{\eta }\right)=\left(1,\sqrt{\frac{b_{n-1}}{c_{n-1}}},\sqrt{\frac{b_{n-1}b_{n-2}}{c_{n-1}c_{n-2}}},\ldots ,\sqrt{\frac{b_{n-1}b_{n-2}\ldots b_{1}}{c_{n-1}c_{n-2}\ldots c_{1}}}\right)$$
we find that, if the products $b_{i}c_{i}$ are positive, then we can define a matrix
(4)$$K=\eta^{\frac{1}{2}}A\eta^{-\frac{1}{2}}=\eta^{-\frac{1}{2}}\left(\eta A\eta^{-1}\right)\eta^{\frac{1}{2}}=\eta^{-\frac{1}{2}}A^{†}\eta^{\frac{1}{2}}=K^{†}$$
which is Hermitian and iso-spectral with *A* and whose diagonal is the same of *A*, while the sub-diagonals are the geometrical means $\left(\sqrt{b_{n-1}c_{n-1}},\ldots ,\sqrt{b_{1}c_{1}}\right)$.

Considering now the distribution
(5)$$h_{\nu }\left(y\right)=\frac{2\exp\left(-y^{2}/2\right)y^{\nu -1}}{2^{\nu /2}\Gamma \left[\nu /2\right]}$$
with a first moment $<y>=\frac{\sqrt{2}\Gamma \left[\left(\nu +1\right)/2\right]}{\Gamma (\nu /2)}∼\sqrt{\nu }$ and a second moment $<y^{2}>=\nu ,$ it is found that by constructing a new variable $z=\sqrt{xy},$ where *x* and *y* are both sorted, its distribution is the *K*-distribution
(6)$$g_{\nu }\left(z=\sqrt{xy}\right)=\frac{8}{2^{\nu }\Gamma^{2}\left(\nu /2\right)}z^{2\nu -1}K_{0}\left(z^{2}\right),$$
where $K_{0}\left(x\right)$ is a modified Bessel function of the second kind. For large values of ν, the contributions to the *K*-distribution come from large values of its argument. This being the case, we are allowed to replace in the expression of $g_{\nu }\left(z\right)$ the Bessel function with its asymptotic expression $K_{0}\left(z\right)∼\sqrt{\frac{\pi }{2z}}\exp\left(-z\right)$, which leads to
(7)$$g_{\nu }\left(z\right)∼\frac{8\sqrt{\pi }}{2^{\nu +1/2}\Gamma^{2}\left(\nu /2\right)}z^{2\nu -2}\exp\left(-z^{2}\right).$$

Using now the gamma function duplication formula
(8)$$\Gamma \left(\mu -\frac{1}{2}\right)=\frac{2^{\mu -1}}{\sqrt{2\pi }}\Gamma \left(\frac{\mu }{2}-\frac{1}{4}\right)\Gamma \left(\frac{\mu }{2}+\frac{1}{4}\right)$$
and the fact that, for large ν,
(9)$$\Gamma \left(\frac{\mu }{2}-\frac{1}{4}\right)\Gamma \left(\frac{\mu }{2}+\frac{1}{4}\right)∼\Gamma^{2}\left(\frac{\mu }{2}\right),$$
it is found that the distribution of the geometrical mean variable can be approximated as
(10)$$g_{\nu }\left(z\right)∼\frac{2}{\Gamma \left(\nu -1/2\right)}z^{2\nu -2}\exp\left(-z^{2}\right)=f_{2\nu -1}\left(z\right)∼f_{2\nu }\left(z\right),$$
where
(11)$$f_{\nu }\left(y\right)=h_{\nu }\left(y\sqrt{2}\right)\sqrt{2}=\frac{2y^{\nu -1}}{\Gamma (\nu /2)}\exp\left(-y^{2}\right).$$

Let us now consider the beta distribution
(12)$$B\left(s,t\right)=\frac{2^{1-s-t}\Gamma \left(s+t\right)}{\Gamma (s)\Gamma (t)}{\left(1-x\right)}^{s-1}{\left(1+x\right)}^{t-1}$$
in the asymptotic situation in which s>>1 and t>>1. In this case, we can make the expansion
(13)$$F\left(x\right)=\left(s-1\right)\log\left(1-x\right)+\left(t-1\right)\log\left(1+x\right)=F\left(x_{s}\right)+F^{\prime }\left(x_{s}\right)\left(x-x_{s}\right)+\frac{1}{2!}F^{\prime \prime }\left(x_{s}\right){\left(x-x_{s}\right)}^{2}\ldots$$
neglecting higher-order terms. Imposing then that $F^{\prime }\left(x_{s}\right)=0,$ after approximating the gamma functions by their Stirling expressions, it is found that the beta distribution approaches the normal distribution
(14)$$B\left(s,t\right)∼N\left(\frac{t-s}{s+t-2},\frac{4(t-1)(s-1)}{{\left(t+s-2\right)}^{3}}\right).$$

For normal distributions, by constructing the mean quantity
(15)$$y=1+\frac{1}{k}\sum_{i=1}^{k}\left(\pm \right)x_{i}$$
in which $x_{i}$ is Gaussian distributed with an average of ${\bar{x}}_{i}$ and variances of $\sigma_{i},$ it is then found that *y* is distributed according to the normal distribution
(16)$$N\left(1+\frac{1}{k}\sum_{i=1}^{k}\left(\pm \right){\bar{x}}_{i},\frac{1}{k^{2}}\sum_{i=1}^{k}\sigma_{i}^{2}\right).$$

## 3. The Pseudo-Hermitian Hermite–β Ensemble

A matrix of the Hermite–β ensemble is a Hermitian tridiagonal matrix whose joint density distribution of the matrix elements is given by
(17)$$P\left(H_{\beta }\right)=\frac{2^{n-1}}{{\left(2\pi \right)}^{n/2}\prod_{j=1}^{n-1}\Gamma \left[\left(N-j\right)\beta /2\right]}\exp\left(-\frac{1}{2}\mathrm{tr}H_{\beta }^{2}\right)\prod_{j=1}^{n-1}b_{j}^{(n-j)\beta -1}$$
such that the diagonal elements $a_{i}$ are normally distributed, namely N(0,1), while the off-diagonal $b_{j}$ elements are distrtributed according to $f_{\nu }\left(y\right),$ as seen in Equation (Equation 11), where $\nu_{j}=\left(n-j\right)\beta$ and β is a real positive parameter. Using the lemmas of the tridiagonal form that relate the Vandermonde determinant to the off-diagonal elements, it is then found that the joint density distribution of the eigenvalues is given by [3]
(18)$$P\left(\lambda_{1},\lambda_{2},\ldots ,\lambda_{n}\right)=C_{n}\exp\left(-\frac{1}{2}\sum_{k=1}^{n}\lambda_{k}^{2}\right)\prod_{j>i}{\left|\lambda_{j}-\lambda_{i}\right|}^{\beta }.$$

By expanding the determinant of the characteristic polynomial, it is obtained that its average satisfies the recursion relationship
(19)$$<P_{n}\left(\lambda \right)>=-\lambda <P_{n-1}\left(\lambda \right)>-\frac{n-1}{2}\beta <P_{n-2}\left(\lambda \right)>$$
such that a comparison with Hermite polynomials recursion relationships leads to the identity $<P_{n}\left(\lambda \right)>={\left(\frac{\beta }{2}\right)}^{\frac{n}{2}}H_{n}\left(\frac{\lambda }{\sqrt{2\beta }}\right).$ This means that, asymptotically, the eigenvalues occupy the same compact support of the zeros of Hermite polynomials [18] defined by Wigner’s semi-circle law [2],
(20)$$\rho_{W}\left(\beta ,\lambda \right)=\frac{1}{\pi \beta }\sqrt{2n\beta -\lambda^{2}}.$$

Considering the fluctuations, we assume that the nearest-neighbor distribution (NND) can be described by the NND of the 2×2 matrices that is found to be given by the generalized Wigner surmise
(21)$$p_{W}\left(\beta ,s\right)=\frac{2\kappa^{\beta +1}s^{\beta }}{\Gamma (\frac{\beta +1}{2})}\exp\left(-\kappa^{2}s^{2}\right),$$
where $\kappa =\Gamma \left(\frac{\beta }{2}+1\right)/\Gamma \left(\frac{\beta +1}{2}\right).$

Let us now remove the Hermitian condition of $H_{\beta }$ by filling one of the off-diagonals by new random variables $c_{j}.$ In [10], this was carried out by sorting the elements of the two sub-diagonals from the same $f_{\nu }\left(y\right)$ distribution of the Hermitian case. However, this choice does not keep the structure of the above joint distribution of elements in terms of the exponential of the trace. If we sort them instead from $h_{\nu }\left(y\right),$ using Equation (Equation 5), the joint distribution of the non-Hermitian matrices ${\hat{H}}_{\beta }$ can be written as
(22)$$P\left({\hat{H}}_{\beta }\right)=\frac{2^{2n-2}}{{\left(2\pi \right)}^{n/2}\prod_{j=1}^{n-1}\Gamma^{2}\left[\left(n-j\right)\beta /2\right]}\exp\left(-\frac{1}{2}\mathrm{tr}{{\hat{H}}_{\beta }}^{2}\right)\prod_{j=1}^{n-1}{\left(b_{j}c_{j}\right)}^{(n-j)\beta -1}.$$

In this case, the recursion relationship of the average characteristic polynomial becomes
(23)$$<{\hat{P}}_{n}\left(\lambda \right)>=-\lambda <{\hat{P}}_{n-1}\left(\lambda \right)>-\left(n-1\right)\beta <{\hat{P}}_{n-2}\left(\lambda \right)>,$$
which leads to the semi-circle law $\rho_{W}\left(2\beta ,\lambda \right),$ in which β is replaced by 2β.

From the preliminary results stated above, ${\hat{H}}_{\beta }$ is a pseudo-Hermitian matrix whose eigenvalues are real, and, at the same time, they also are eigenvalues of a Hermitian matrix $K_{\beta }.$ The diagonal elements of $K_{\beta }$ are the same of ${\hat{H}}_{\beta }$, while the off-diagonal elements are the geometrical mean $d_{i}=\sqrt{b_{i}c_{i}}$ of the ${\hat{H}}_{\beta }$ elements and are therefore distributed according to Equation (Equation 6). Using, then, the asymptotic of these functions and Equation (Equation 7), the joint density distribution of the $K_{\beta }$ approaches the distribution
(24)$$P\left(K_{\beta }\right)∼\frac{2^{n-1}}{{\left(2\pi \right)}^{n/2}\prod_{j=1}^{n-1}\Gamma \left[\left(n-j\right)\beta \right]}\exp\left(-\frac{1}{2}\mathrm{tr}K_{\beta }^{2}\right)\prod_{j=1}^{n-1},d_{j}^{(n-j)2\beta -1},$$
which is just the joint distribution of the Hermitian case with a β replaced by 2β. The joint distribution of the eigenvalues is then given by Equation (Equation 18), with β replaced by 2β. In Figure 1, this 2β effect is illustrated in the case of the eigenvalues of the pseudo-Hermitian β–Hermite ensemble.

## 4. The Pseudo-Hermitian Laguerre–β Ensemble

Let *B* be the bidiagonal matrix with diagonal elements $z=(z_{n},z_{n-1},\ldots ,z_{1})$ and sub-diagonal elements $x=(x_{n-1},x_{n-2},\ldots ,x_{1})$, whose joint distribution of the elements is given by
(25)$$P\left(B_{\beta }\right)\propto \prod_{i=1}^{n}z_{n-i+1}^{2\alpha -(i-1)\beta -1}\exp\left(-\frac{1}{2}z_{n-i+1}^{2}\right)\prod_{j=1}^{n-1}x_{n-j}^{(n-j)\beta -1}\exp\left(-\frac{1}{2}x_{n-j}^{2}\right),$$
where α=mβ/2 with m≥n. Therefore, the elements obey the distribution $h_{\nu }\left(y\right),$ where ν=2α−(i−1)β for the diagonal elements and ν=(n−j)β for the sub-diagonal elements. The Laguerre β ensemble is then defined as the tridiagonal matrix
(26)$$L_{\beta }=BB^{T}=\left(\begin{array}{cccccc} z_{n}^{2} & z_{n}x_{n-1} & \; & & \; & \\ z_{n}x_{n-1} & z_{n-1}^{2}+x_{n-1}^{2} & z_{n-1}x_{n-2} & \; & & \\ \; & . & . & . & \; & \\ \; & \; & . & . & . & \\ \; & \; & & . & . & .\\ \; & \; & & z_{3}x_{2} & z_{2}^{2}+x_{2}^{2} & z_{2}x_{1}\\ \; & \; & & \; & z_{2}x_{1} & z_{1}^{2}+x_{1}^{2} \end{array}\right).$$

As shown in Ref. [3], in order to change from a bidiagonal to a tridiagonal matrix, the Jacobian
(27)$$J_{B\to L}={\left(2^{n}z_{1}\prod_{i=2}^{n}z_{i}^{2}\right)}^{-1}$$
is used to give
(28)$$P\left(L_{\beta }\right)\propto 2^{-n}z_{1}^{2\alpha -(n-1)\beta -2}\exp\left(-\frac{1}{2}\mathrm{tr}L_{\beta }\right)\prod_{i=1}^{n-1}z_{i+1}^{2\alpha -(n-1-i)\beta -3}\prod_{j=1}^{n-1}x_{j}^{j\beta -1}$$
such that, using lemmas of the tridiagonal form, it is found that the joint distribution of the eigenvalues is given by [3]
(29)$$P\left(\lambda_{1},\lambda_{2},\ldots ,\lambda_{n}\right)=C_{n,m}\exp\left(-\frac{1}{2}\sum_{i=1}^{n}\lambda_{i}\right)\prod_{i=1}^{n}\lambda_{i}^{\left(m-n+1\right)\frac{\beta }{2}-1}\prod_{i\neq j}{\left|\lambda_{j}-\lambda_{i}\right|}^{\beta }.$$

Let us start by obtaining the average characteristic polynomial whose derivation is also valid in the pseudo-Hermitian case. By expanding the determinant polynomial and taking the average of elements, we have the recursion relationship
(30)$$<P_{n}\left(\lambda \right)>=\left(m\beta -\lambda \right)<R_{n-1}\left(\lambda \right)>-m\left(n-1\right)\beta^{2}<R_{n-2}\left(\lambda \right)>,$$
where the *R*-polynomials satisfy the recursion relationship
(31)$$<R_{n-1}\left(\lambda \right)>=\left[(m+n-2)\beta -\lambda \right]<R_{n-2}\left(\lambda \right)>-\left(m-1\right)\left(n-2\right)\beta^{2}<R_{n-3}\left(\lambda \right)>.$$

These above relationships, respectively, are solved by the two generalized Laguerre polynomials $<R_{n-1}\left(\lambda \right)>=\left(n-1\right)!\beta^{n-1}L_{n-1}^{m-n+1}\left(\frac{\lambda }{\beta }\right)$ and $<P_{n}\left(\lambda \right)>=\left(n\right)!\beta^{n}L_{n}^{m-n}\left(\frac{\lambda }{\beta }\right).$ The density of the eigenvalues, which is also that of the zeros of the polynomials, is given by the Marchenko–Pastur expression
(32)$$\rho \left(\lambda \right)=\frac{1}{2\beta \pi \lambda }\sqrt{\left(\lambda_{+}-\lambda \right)\left(\lambda -\lambda_{-}\right)},$$
where $\lambda_{\pm }=n\beta {\left(\sqrt{\frac{m}{n}}\pm 1\right)}^{2}$, and from this, we have the cumulative function
(33)$$N\left(\lambda \right)=\frac{1}{4\pi \beta }\left\{\begin{array}{cc} -4\sqrt{\lambda_{+}\lambda_{-}}\arctan\sqrt{\frac{\lambda_{+}\left(\lambda -\lambda_{-}\right)}{\lambda_{-}\left(\lambda_{+}-\lambda \right)}}+\left(\lambda_{+}+\lambda_{-}\right)\arccos\frac{\lambda_{+}+\lambda_{-}-2\lambda }{\lambda_{+}-\lambda_{-}} & \\ +2\sqrt{\left(\lambda_{+}-\lambda \right)\left(\lambda -\lambda_{-}\right)} & \end{array}.\right.$$

To construct the pseudo-Hermitian–Laguerre ensemble, we introduce a new bidiagonal matrix *C* with a diagonal $w=(w_{n},w_{n-1},\ldots ,w_{1}),$ and a lower sub-diagonal $y=(y_{n-1},y_{n-2},\ldots ,y_{1}),$ whose elements are sorted from the same distribution of the elements of *B*. Once this is carried out, a non-Hermitian β–Laguerre matrix is obtained by taking the product ${\hat{L}}_{\beta }=BC^{T}$ that produces the matrix
(34)$${\hat{L}}_{\beta }=\left(\begin{array}{cccccc} z_{n}w_{n} & z_{n}y_{n-1} & \; & & \; & \\ w_{n}x_{n-1} & z_{n-1}w_{n-1}+x_{n-1}y_{n-1} & z_{n-1}y_{n-2} & \; & & \\ \; & w_{m-1}x_{m-2} & . & . & \; & \\ \; & \; & . & . & . & \\ \; & \; & & . & . & .\\ \; & \; & & w_{3}x_{2} & z_{2}w_{2}+x_{2}y_{2} & z_{2}y_{1}\\ \; & \; & & \; & w_{2}x_{1} & z_{1}w_{1}+x_{1}y_{1} \end{array}\right)$$

Immediately, the pseudo-Hermitian nature of this matrix is proved, and, furthermore, it can be seen that it is iso-spectral with the Hermitian matrix $M_{\beta }=DD^{T}$ in which *D* is the bidiagonal matrix
(35)$$D=\left(\begin{array}{cccccc} \sqrt{z_{n}w_{n}} & \; & & \; & & \\ \sqrt{x_{n-1}y_{n-1}} & \sqrt{z_{n-1}w_{n-1}} & \; & & \; & \\ \; & \sqrt{x_{n-2}y_{n-2}} & \sqrt{z_{n-2}w_{n-2}} & . & \; & \\ \; & \; & . & . & . & \\ \; & \; & & . & . & .\\ \; & \; & & \sqrt{x_{2}y_{2}} & \sqrt{z_{2}w_{2}} & \\ \; & \; & & \; & \sqrt{x_{1}y_{1}} & \sqrt{z_{1}w_{1}} \end{array}\right).$$

Denoting the diagonal elements as $a_{i}=\sqrt{z_{i}w_{i}}$ and the sub-diagonals as $b_{i}=\sqrt{x_{i}y_{i}},$ of course, they are distributed according to $g_{\nu }\left(y\right)$, as per Equation (Equation 6), with ν=2α−(j−1)β for the diagonals and ν=(n−j)β for the sub-diagonals. Replacing $g_{\nu }\left(y\right)$ with its asymptotic expression, we obtain
(36)$$P\left(M_{\beta }\right)\propto a_{1}^{4\alpha -2(n-1)\beta -2}\exp\left(-\mathrm{tr}M_{\beta }\right)\prod_{i=1}^{n-1}a_{i+1}^{4\alpha -2(n-1-j)\beta -3}\prod_{j=1}^{n-1}b_{j}^{2j\beta -1},$$
and for the eigenvalues, we obtain
(37)$$P\left(\lambda_{1},\lambda_{2},\ldots ,\lambda_{n}\right)=C_{n,m}\exp\left(-\sum_{i=1}^{n}\lambda_{i}\right)\prod_{i=1}^{n}\lambda_{i}^{(m-n+1)\beta -1}\prod_{i\neq j}{\left|\lambda_{j}-\lambda_{i}\right|}^{2\beta }.$$

In Figure 1, the 2β effect is illustrated in the case of the pseudo-Hermitian β–Laguerre ensemble.

## 5. The Pseudo-Hermitian Jacobi–β Ensemble

A matrix of the Jacobi β ensemble is a Hermitian tridiagonal matrix [4,5,6,19]
(38)$$J_{\beta }=\left(\begin{array}{cccccc} a_{1} & b_{1} & \; & & \; & \\ b_{1} & a_{2} & b_{2} & \; & & \\ \; & . & . & . & \; & \\ \; & \; & . & . & . & \\ \; & \; & & . & . & .\\ \; & \; & & b_{n-2} & a_{n-1} & b_{n-1}\\ \; & \; & & \; & b_{n-1} & a_{n} \end{array}\right)$$
where the diagonal elements are given by
(39)$$a_{k+1}=\left(1-\alpha_{2k-1}\right)\alpha_{2k}-\left(1+\alpha_{2k-1}\right)\alpha_{2k-2}$$
and the off-diagonal elements are given by
(40)$$b_{k+1}=\sqrt{\left(1-\alpha_{2k-1}\right)\left(1-\alpha_{2k}^{2}\right)\left(1+\alpha_{2k+1}\right)},$$
where, with 0≤k≤2n−1, the $\alpha_{k}$ are β-distributed, following Equation (Equation 12), with variables given by
(41)$$B\left(\frac{2n-k-2}{4}\beta +a+1,\frac{2n-k-2}{4}\beta +b+1\right)\mathrm{for}\;k\;\mathrm{even}$$
and
(42)$$B\left(\frac{2n-k-3}{4}\beta +a+b+2,\frac{2n-k-1}{4}\beta \right)\mathrm{for}\;k\;\mathrm{odd},$$
aside from the special cases $\alpha_{2n-1}=\alpha_{-1}=-1.$ It has been proven that the joint distribution of the eigenvalues of the these matrices is given by
(43)$$P\left(\lambda_{1},\lambda_{2},\ldots ,\lambda_{n}\right)=C_{n}\prod_{i=1}^{n}{\left(2-\lambda_{i}\right)}^{a-1}{\left(2+\lambda_{i}\right)}^{b-1}\prod_{j>i}{\left|\lambda_{j}-\lambda_{i}\right|}^{\beta }.$$

The density of the eigenvalues satisfies an inverted semi-circle law
(44)$$\rho \left(\lambda \right)=\frac{n}{\pi \sqrt{4-\lambda^{2}}}+c,$$
where the constant c is a small parameter necessary to adjust the results of the numerical simulations. As a consequence, the unfolding of the eigenvalues is performed with the cumulative function
(45)$$N\left(\lambda \right)=\frac{n}{\pi }\arcsin\left(\frac{\lambda }{2}\right)+c\lambda .$$

As the above parameters of the beta distributions increase for large matrix sizes, it is justified to assume that when n>>1, they can be replaced by the Gaussian approximations
(46)$$N\left(\frac{2(b-a)}{(2n-k-2)\beta +2a+2b},\frac{2[(2n-k-2)\beta +4a)][(2n-k-2)\beta +4b]}{{\left[\left(2n-k-2\right)\beta +2a+2b\right]}^{3}}\right)\mathrm{for}\;k\;\mathrm{even}$$
and
(47)$$N\left(\frac{\beta -2(a+b)-4}{(2n-k-2)\beta +2a+2b},\frac{2[(2n-k-3)\beta +4a+4b+4)][(2n-k-1)\beta -4]}{{\left[\left(2n-k-2\right)\beta +2a+2b\right]}^{3}}\right)\mathrm{for}\;k\;\mathrm{odd},$$
which were derived in the preliminary section.

The important point to observe in these equations is that the average positions and the variances of the above Gaussians approach zero when the size *n* of the matrices becomes arbitrarily large. As a consequence, the above expressions connecting the matrix elements to the variables $\alpha_{k}$ can be linearized as $a_{k+1}∼\alpha_{2k}-\alpha_{2k-2}$ for the diagonal elements and $b_{k+1}∼1-\left(\alpha_{2k-1}-\alpha_{2k+1}\right)/2$ for the off-diagonal ones. These approximated expressions show that, asymptotically, the diagonal and the off-diagonal elements decouple, and, moreover, all elements become Gaussian distributed, such that we have for the diagonal elements
(48)$$P\left(a_{k+1}\right)∼N\left({\bar{\alpha }}_{2k}-{\bar{\alpha }}_{2k-2},\sigma_{2k}^{2}+\sigma_{2k-2}^{2}\right),$$
and for the off-diagonal elements, we have
(49)$$P\left(b_{k+1}\right)∼N\left(1-\frac{{\bar{\alpha }}_{2k-1}-{\bar{\alpha }}_{2k+1}}{2},\frac{\sigma_{2k-1}^{2}+\sigma_{2k+1}^{2}}{4}\right).$$

Therefore, the off-diagonal elements fluctuate around one, while the diagonal ones fluctuate around zero. Thus, it is reasonable to expect that the off-diagonal elements have a determinant role in the behavior of the eigenvalue properties inasmuch as, for tridiagonal matrices, the eigenvalues are directly connected to the off-diagonal elements. The precision of this approximated description as compared to the exact expressions is attested to in Figure 2.

Therefore, the matrix elements are constructed using the 2n−1 random components of the vector $\alpha \left(k\right)$, which are shared by both the diagonal and the sub-diagonal elements. Thus, it does not seem to have a clear way to remove the Hermitian condition. On the other hand, the basic idea of the 2β Dyson effect is that it is caused by doubling the number of independent variables, especially those in the off-diagonal matrix elements. Considering now the removal of the Hermitian condition of the matrices, this can be executed by just sorting a new vector $\tilde{\alpha }\left(k\right)$ from which only the odd elements are used to fill the lower sub-diagonal. Therefore, only the number of odd elements are doubled. This defines a lower sub-diagonal whose elements $c_{k+1}$ differ from $b_{k+1}$ by odd terms. As the elements of these two off-diagonal are positive, these non-Hermitian matrices, besides being pseudo-Hermitian, are iso-spectral with the Hermitian matrix whose off-diagonal elements are given by $\sqrt{b_{k+1}c_{k+1}}.$

Turning now to the asymptotic description, consistent with our level of approximations, the geometrical mean of the two off-diagonal elements can be linearized as
(50)$$\sqrt{b_{k+1}c_{k+1}}∼1-\left[\left(\alpha_{2k-1}-\alpha_{2k+1}\right)+\left(\alpha_{2k-1}^{\prime }-\alpha_{2k+1}^{\prime }\right)\right]/4,$$
where the primes denote α variables belonging to the $c_{k+1}$ sub-diagonal. Using the result (Equation 16) from Section 2, we can write
(51)$$P\left(\sqrt{b_{k+1}c_{k+1}}\right)∼N\left(1-\frac{{\bar{\alpha }}_{2k-1}-{\bar{\alpha }}_{2k+1}}{2},\frac{1}{2}\frac{\sigma_{2k-1}^{2}+\sigma_{2k+1}^{2}}{4}\right)$$
such that the factor 1/2 affects the variances of the odd *k* terms as
(52)$$\frac{2[(2n-k-3)2\beta +8(a+b+1)][(2n-k-1)2\beta -8]}{{\left[\left(2n-k-2\right)2\beta +2\left(a+b\right)\right]}^{3}}.$$

The above expression shows the occurrence of a 2β effect, which affects only the odd terms of the off-diagonal elements. This means that we are dealing with a hybrid pseudo-Hermitian ensemble that combines the even elements generated with β and the odd ones sorted with 2β. The evidence from numerical simulations (see Figure 3) suggests that the spectral statistics, that is, the NNSD, can be fitted with a parameter $\sqrt{2}\beta .$ It is interesting to observe that $\sqrt{2}\beta$ is the geometric mean of β and 2β. To ’fix’ this, what we can do is, in the pseudo-Hermitian case, generate the even elements with a parameter 2β while sorting the odd elements with β. The results of the numerical simulations following this procedure demonstrate a clear 2β effect, as is shown in Figure 4 for the NNSD and in Figure 5 for the number variance.

## 6. Conclusions

In conclusion, we have extended the universality of RMT Wigner–Dyson statistics to a class of non-Hermitian matrices with real eigenvalues. Matrices of this class are connected to their adjoints by a similarity transformation, a discrete symmetry condition that defines a pseudo-Hermitian matrix. In our case, the matrices are constructed by removing the Hermitian condition of the tridiagonal matrices of the three so-called β ensembles. By doing this, the number of off-diagonal independent elements doubles, and, as also occurs in standard random matrix theory, the Dyson β index also doubles. One obvious question is to investigate if this 2β effect can be observed in pseudo-Hermitian full random matrices. However, it is not clear how, in this case, a pseudo-Hermitian model can be constructed in which the number of independent variables doubles. Another question is if the effect can be observed in a Hamiltonian with real eigenvalues, as this can lead to a link between *PT* symmetry and quantum chaos. As a final remark, we emphasize that the origin of the 2β effect stems from the asymptotic behavior of the geometric mean calculated with the distribution functions of the matrix elements of the three ensembles. This suggests another universality associated with random matrix theories.

## Figures and Tables

**Figure 1 entropy-25-00868-f001:**
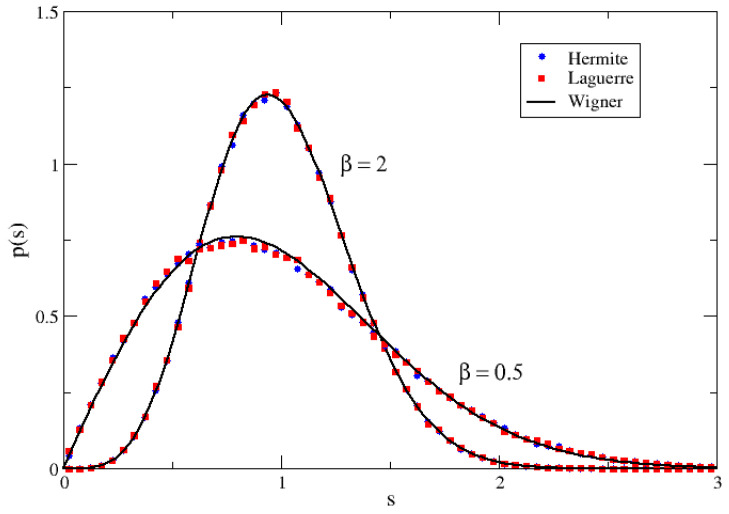
NNSD for pseudo-Hermitian matrices of size *n* = 128 of the Hermite (blue circles) and Laguerre ensembles (red squares) for the indicated values of β=0.5 and β=2. The black full lines were calculated with Wigner surmise and Equation (Equation 21), with β=1 and β=4, respectively.

**Figure 2 entropy-25-00868-f002:**
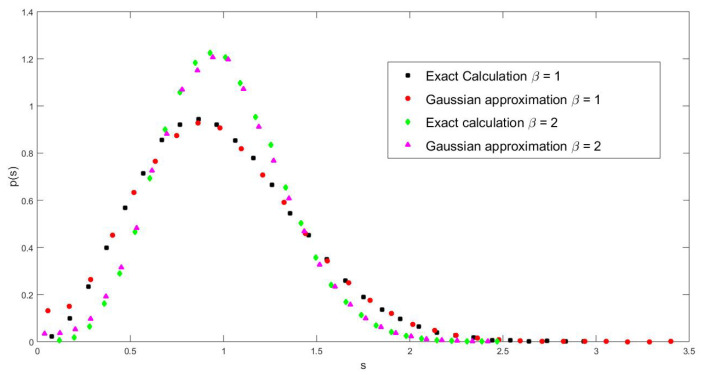
Comparison of the Gaussian approximation with the exact calculation of NNSD for β=1 and β=2, as indicated in the figure.The calculations were performed with 50 matrices of size *n* = 450.

**Figure 3 entropy-25-00868-f003:**
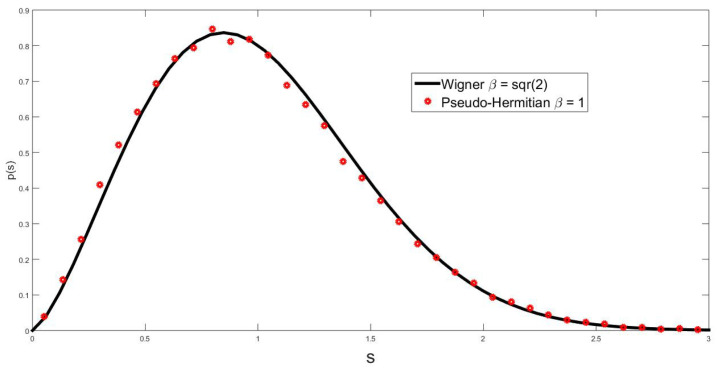
The hybrid pseudo-Hermitian β–Jacobi ensemble in which even elements are sorted with β, while the odd ones, asymptotically, become sorted with 2β. The blue line is calculated with Equation (Equation 21) using as parameters the geometric mean of β and 2β, that is, $\sqrt{2}\beta$. The calculations were performed with 50 matrices of size *n* = 360.

**Figure 4 entropy-25-00868-f004:**
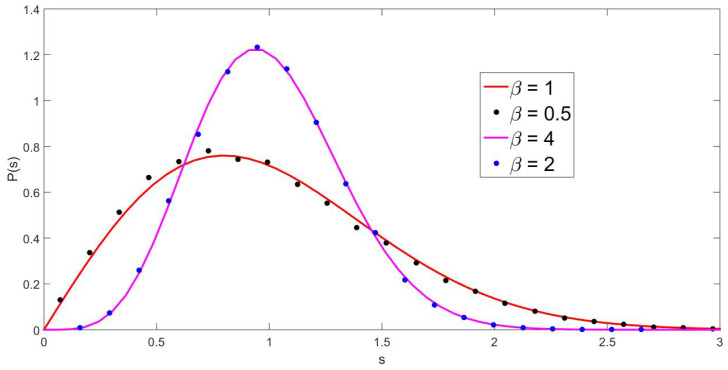
NNSD for 100 matrices of size 120 in which the even elements of the pseudo-Hermitian β–Jacobi ensemble are sorted with 2β (black and blue dots) for the indicated values of beta. The full lines were calculated with Wigner surmise, as seen in Equation (Equation 21), with β=1 and β=4, respectively.

**Figure 5 entropy-25-00868-f005:**
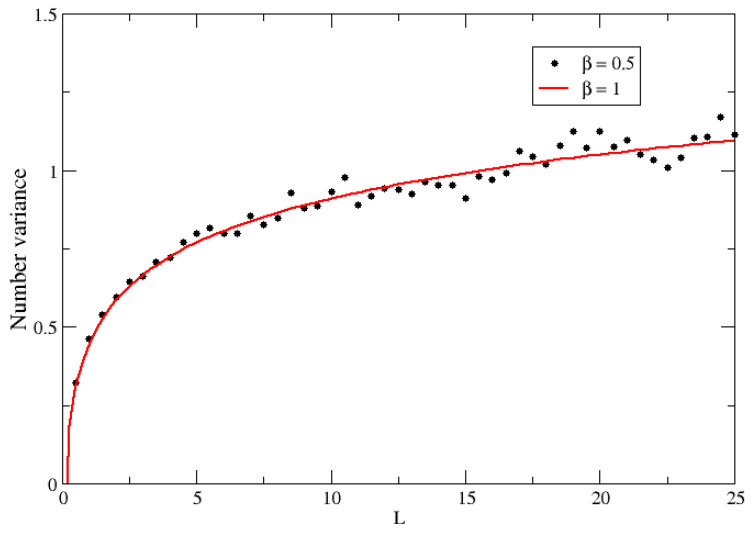
The verification of the 2β effect in the number variance statistics: black dots were calculated with 600 matrices of size *n* = 120 of the pseudo-Hermitian β–Jacobi ensemble sorted with β=0.5, and the red full line corresponds to the GOE number variance.

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
