# Peer review of "The Double Dyson Index β Effect in Non-Hermitian Tridiagonal Matrices"

_entropy, 2023, doi:10.3390/e25060868_

Round 1
Reviewer 1 Report
The main observation is mathematically interesting, which is that a non-Hermitian matrix ensemble, of 'Dumitriu-Edelman' type, is pseudo-Hermitian. After conjugation to make it Hermitian, it is probabilistically similar to something similar to the usual Dumitriu-Edelman, but with different noise scaling. Hence it lands in the Dyson-Mehta universality class.
However, there is a major mathematical mistake in matching parameters. It is NOT correct that for Hermitian tridiagonal matrices, the Dyson-index only depends on the variance of the off-diagonal entries. The authors should try the following to convince themselves:
1) Generate the Dumitriu-Edelman matrix A. Look at its empirical spectral spacing properly inside the bulk.
2) Generate the Dumitriu-Edelman matrix A, but then set its diagonal entries to 0. Look at it's empirical spectral spacing properly inside the bulk. This will follow a different Wigner Surmise. This is IN SPITE of having the same semicircle law and IN SPITE of having the same Hermite polynomial as the expected characteristic polynomial.
After scaling the Dumitriu-Edelman-like matrix so that the semicircle will be on [-1,1], it is the sum of the variances of diagonal and off-diagonal which determines the Dyson index.
Reviewer 2 Report
Please see the attached report.

Round 2
Reviewer 1 Report
The authors claim in their rebuttal:
"However, with regard to the mathematical flaw appointed by him, we think that there is a misunderstanding: nowhere in the article it is said that the variance of the off-diagonal elements determine the value of the Dyson beta index"
Here is a non-exhaustive list of places where they are either imprecise in their wording or claim exactly this point.
1. In the abstract:
"Our purpose, however is to show that, when the Hermitian condition of the real matrices generated with a given value of β, is removed, and, as a consequence, the number of non-diagonal independent variables doubles, non-Hermitian matrices exist that asymptotically behave as if they had been generated with a value 2β"
It's not simply a question of the variance or 'number' of offdiagonals, and yet here it is claimed the Dyson index doubles. The construction I gave also 'doubles' the number of off-diagonal variables, but will have different Dyson index than they claim.
2. "Therefore, the off-diagonal elements fluctuate around one, while the diagonal ones fluctuate around zero, thus, it is reasonable to expect that the off-diagonal elements have a determinant role in the behavior of the eigenvalue properties. Inasmuch as, for tridiagonal matrices, the eigenvalues are directly connected to the off-diagonal elements."
Sorry. This is completely wrong. Please consider the example I gave.
3. In the conclusion, "By doing this, the number of off-diagonal independent elements doubles and, as it happens in standard random matrix theory, Dyson β index also doubles."
Sorry this is just wrong. I gave you a counterexample, and this is absolutely not 'standard'. Having put the support of the semicircle on (-1,1), the Dyson index will be determined by the sum of squares of variances of diagonal and off-diagonal.
Your simulations seem to show the conclusions are correct, so it is just the explanation which is off.
Author Response
Response to reviwer1
We believe that beyond the misunderstanding there is an agreement between us and the referee. As it is stated in the Conclusions, we are extending the universality of the Wigner-Dyson statistics to non-Hermitian ensembles. That is the merit of our work.
Below we put the referee quotes of our article then his(her) comments and after our responses:
1."Our purpose, however, is to show that, when the Hermitian condition of the real matrices generated with a given value of β, is removed, and, as a consequence, the number of non-diagonal independent variables doubles, non-Hermitian matrices exist that asymptotically behave as if they had been generated with a value 2β"
It's not simply a question of the variance or 'number' of offdiagonals, and yet here it is claimed the Dyson index doubles. The construction I gave also 'doubles' the number of off-diagonal variables, but will have different Dyson index than they claim.
It seems that he(she) is missing the “non-Hermitian matrices exist” we obviously are not claiming that Dyson index autommatically doubles. Again, we do not need the referee’s construction as it is said in the introduction that the previous non-Hermitian construction of Ref. [10] did not produce the 2-beta effect. We call the attention of the referee for the paragraph after eq. (21) that explicitly states that the effect does not come unless the distribution of the off-diagonal elements is modified.
- "Therefore, the off-diagonal elements fluctuate around one, while the diagonal ones fluctuate aroundzero, thus, it is reasonable to expect that the off-diagonal elements have a determinant role in the behavior of the eigenvalue properties. Inasmuch as, for tridiagonal matrices, the eigenvalues are directly connected to the off-diagonal elements."
Sorry. This is completely wrong. Please consider the example I gave.
We are just referring to the famous lemma that relates the Vandermond determinant
of the eigenvalues to the off-diagonal elements of tridiagonal matrices. Please, look at the Dumitriu & Edelman article.
- In the conclusion, "By doing this, the number of off-diagonal independent elementsdoubles and, as it happens in standard random matrix theory, Dyson β index also doubles."
Sorry this is just wrong. I gave you a counterexample, and this is absolutely not 'standard'. Having put the support of the semicircle on (-1,1), the Dyson index will be determined by the sum of squares of variances of diagonal and off-diagonal.
Of course, we are talking about the classes of the standard random matrix theory in which going from real to complex elements and from complex to quaternion elements, the Dyson index goes from 1 to 2 and from 2 to 4, that is, it doubles. We do not understand the importance the referee gives to “the semicircle on (-1,1)” which is just a rescaling of the eigenvalue variables and cannot by itself prove or disprove anything.
Reviewer 2 Report
The authors have successfully addressed the points raised on the previous manuscript, except the point 7 regarding the figure 4. Probably I should have formulated my sentence more clearly. Please note that there are two Wigner surmise curves in this plot, one for β=1 and the other for β=4. However, both are in red color and therefore, at a glance, a general reader may not distinguish one from the other. The corresponding dots are also of the same kind (black for both cases). What I meant in my earlier report by "Moreover, it is advisable to use different symbols for β=0.5 and β=1 cases" was that if instead of dots for both, say triangle is use for one of the cases, then it would be easier to distinguish them (and in this case, changing the color of Wigner surmise curves may not be necessary). Also, please do use the symbol 'β' in the legends instead of 'beta'.
Overall, I recommend the publication of this manuscript. Figure 4 may be updated during the production process. Further review is not required from my side.
Author Response
The Fig. 4 was redone as requested by the referee.